# Glucose Tolerance Test and Pharmacokinetic Study of *Kaempferia parviflora* Extract in Healthy Subjects

**DOI:** 10.3390/nu11051176

**Published:** 2019-05-25

**Authors:** Bungorn Sripanidkulchai, Catheleeya Mekjaruskul, Rosawan Areemit, Areewan Cheawchanwattana, Jiraporn Sithithaworn

**Affiliations:** 1Center for Research and Development of Herbal Health Products, Faculty of Pharmaceutical Sciences, Khon Kaen University, Khon Kaen 40002, Thailand; 2Faculty of Pharmacy, Mahasarakham University, Maha Sarakham 44150, Thailand; catheleeya.m@msu.ac.th; 3Faculty of Medicine, Department of Pediatrics, Khon Kaen University, Khon Kaen 40002, Thailand; rosawan@kku.ac.th; 4Faculty of Pharmaceutical Sciences, Khon Kaen University, Khon Kaen 40002, Thailand; areche@kku.ac.th; 5Faculty of Medicine, Mahasarakham University, Maha Sarakham 44000, Thailand; jirapo_n@kku.ac.th

**Keywords:** glucose tolerance, pharmacokinetic, *Kaempferia parviflora*, methoxyflavone

## Abstract

*Kaempferia parviflora* Wall. ex Baker (KP), Krachaidam in Thai or Thai ginseng, is a herbal medicine that has many potential pharmacological effects. The effect of KP extract on blood glucose level in rodent was reported. This study focused on the oral glucose tolerance test and pharmacokinetic study in healthy volunteers administered with KP extract (90 and 180 mg/day, placebo). The oral glucose tolerance tests were performed at baselines and 28-days of administration. The pharmacokinetics were determined after a single dose administration of the tested products using 3,5,7,3′,4′-pentamethoxyflavone (PMF) and 5,7,4′-trimethoxylflavone (TMF) as markers. The results showed that glucose metabolism via oral glucose tolerance test was not affected by KP extract. Blood glucose levels of volunteers at 120 min after glucose loading were able to be returned to initial levels in placebo, KP 90 mg/day, and KP 180 mg/day groups both at baseline and 28-days of administration. The results of the pharmacokinetic study revealed that only TMF and PMF, but not 5,7-dimethoxyflavone (DMF) levels could be detected in human blood. The given doses of KP extract at 90 and 180 mg/day showed a linear dose-relationship of blood PMF concentration whereas blood TMF was detected only at high given dose (180 mg/day). The half-lives of PMF and TMF were 2–3 h. The maximum concentration (C_max_), area under the curve of blood concentration and time (AUC), and time to maximum concentration (T_max_) values of PMF and TMF estimated for the 180 mg/day dose were 71.2 ± 11.3, 63.0 ± 18.0 ng/mL; 291.9 ± 48.2, 412.2 ± 203.7 ng∙h/mL; and 4.02 ± 0.37, 6.03 ± 0.96 h, respectively. PMF was quickly eliminated with higher Ke and Cl than TMF at the dose of 180 mg/day of KP extract. In conclusion, the results demonstrated that KP extract had no effect on the glucose tolerance test. In addition, this is the first demonstration of the pharmacokinetic parameters of methoxyflavones of KP extract in healthy volunteers. The data suggest the safety of the KP extract and will be of benefit for further clinical trials using KP extract as food and sport supplements as well as a drug in health product development.

## 1. Introduction

*Kaempferia parviflora* Wall. ex Baker (KP) (Thai ginseng, black ginger, or Krachaidam in Thai) belongs to the family of Zingiberaceae. Several flavonoids are constituents of KP with three major compounds having been used as the markers for quantitative analysis, namely 3,5,7,3′,4′-pentamethoxyflavone (PMF, PubChem CID: 97332), 5,7,4′-trimethoxyflavone (TMF, PubChem CID: 79730), and 5,7-dimethoxyflavone (DMF, PubChem CID: 88881) [1]. Ethnopharmacologically, the plant rhizomes have been traditionally used in folk medicine for centuries for longevity promotion, anti-fatigue, appetite induction, male sexual stimulation, anti-stomachache, and laxative [2,3]. KP extract and its major constituents have been reported to confer several health beneficial effects through in vitro and animal studies, including aphrodisiac activity [4,5,6,7], anti-inflammation [8], anti-cancer [9,10], cardioprotection [11,12], anti-peptic ulcer [13], antimicrobial [14], anti-allergy [15], anti-mutagenicity [16], anti-depression [17], anti-cholinesterase activity [17,18], prevention in the brain from valproic acid-induced impairment of spatial memory [19], inhibition of intrinsic aging processes [20], anti-osteoporosis [21], reduction of pain threshold and severity of osteoarthritis [22], anti-obesity and prevention of obesity-induced dermatopathy [23,24], inhibition of fat accumulation and muscle atrophy [25], increased whole-body energy expenditure [26], improvement of obesity and insulin resistance [27], and promotion of the differentiation of brown adipose tissue [28,29]. In terms of antidiabetic activity, the ethanol extract of KP was reported to decrease blood glucose in streptozocin-induced diabetic rats [30]. Later on, 5,7,3′,4′-tetramethoxyflavone, TMF, and PMF constituents from KP showed in vitro inhibitory effects on α-glucosidase activity [31]. Moreover, the ethyl acetate extract of KP also suppressed the glucose and lipid metabolism in Tsumura, Suzuki, Obese Diabetes (TSOD) mice [23,30]. Several clinical trials showed the potential activities of KP extract including increased physical fitness in elderly volunteers and in soccer players [32,33], decreased abdominal fat in overweight and pre-obese subjects [34], improved self-assessed sex health in men [35], and reduction of stress and anxiety in adult subjects [36]. Moreover, the safety of KP has been reported. A study of acute toxicity in mice found that the LD_50_ value of KP is more than 13.3 g/kg. Oral administration of KP extract at a single dose of 2 g/kg was safe. For chronic toxicity studies, daily doses of KP extract up to 2000 mg/kg given to rats for six months did not show any abnormality in histopathological examination of organs, behaviors, physical examination, and body weight [8,37]. Accordingly, these efficacies and its safety indicate that KP has a potential to be the new product from Thai herbal plants for this century. Although pharmacokinetic data of KP extract in rats have been reported [38], to our knowledge, there are no reports of pharmacokinetics in human. It is necessary to have a better understanding of the pharmacokinetics of this plant in humans. Moreover, there are no reports regarding an oral glucose tolerance test that explain glucose metabolism in humans after administration of KP extract. Therefore, the objectives of this study were to evaluate the effect of a single dose administration of 90 and 180 mg/day of KP extract on the oral glucose tolerance test and its pharmacokinetic parameters in healthy volunteers.

## 2. Materials and Methods

### 2.1. Chemicals and Test Product Description

Glucose powder was purchased from Utopian Co. Ltd., (Samutprakan), Thailand, and 17-α-hydroxyprogesterone was purchased from Sigma, (Shanghai) China. Acetonitrile was analytical grade from RCI Labscan, (Bangkok) Thailand. Methanol was HPLC grade from RCI Labscan, (Bangkok), Thailand.

Dried powder of KP Romkaou strain rhizomes obtained from Phurue, Loei province of Thailand was authenticated and kept as a voucher specimen (No. KP-BS-2010) at the Center for Research and Development of Herbal Health Products, Khon Kaen University, (Khon Kaen), Thailand. KP extract was prepared by maceration in 95% ethanol (following the petty patent of Thailand No. 4048) and the crude extract was obtained at 5.71% yield. The KP extract was analyzed for the content of methoxyflavones using HPLC (Figure 1). The tested KP product was prepared in the dosage form of a capsule to contain 90 mg of KP extract and other excipients (Table 1). The placebo capsule containing no KP extract was composed of the same excipients and capsule color. The entire study was conducted using a single batch of KP extract to optimize product consistency.

### 2.2. Study Design

Randomized, double blinded, and placebo control trials were conducted in 45 healthy volunteers (Figure 2). The sample size was 15 per group, which was calculated by using power of 80%, alpha of 0.05, and standardized effect size of 1.05. The clinical study was approved by the Khon Kaen University Ethics Committee in human research, Thailand (No. 4.2.11:10/2552) following the Declaration of Helsinki and ICH Good Clinical Practice. All volunteers signed the informed consent form before participating in the study. Inclusion criteria included healthy individuals, 20 to 50 years old, fasting plasma glucose between 70 and 100 mg/dL, and normal range of AST, ALT, and serum creatinine levels. Participants who had impaired glucose tolerance test, were allergic to KP, pregnant, lactating, consuming other medicines or herbal medicines, or participating in other trials were excluded from this study. All of the recruited participants were screened for healthy status by a physician. Their blood biochemical parameters and complete blood count (CBC) were determined as the baseline data and after the 28-day study period to monitor the safety of the treatment.

After overnight fasting of 8–12 h, the volunteers were randomly divided to three groups of 15 people; each subject was daily and orally taking KP extract 90 mg/day (one capsule of KP extract and one capsule of placebo), KP extract 180 mg/day (two capsules of KP extract), or placebo (two capsules of placebo) for groups 1, 2, and 3, respectively, for 28 days consecutively. Due to our preliminary study in rats using KP extract 150 and 300 mg/kg body weight was able to decrease blood glucose level in rat after treatment for 4 weeks (unpublished data). The safety factor was 100. Sixty kilograms of body weight was calculated. Therefore, KP extract 90 and 180 mg/kg BW were selected for this study. At baseline and day 28 after taking the capsules, all subjects were given a 20% glucose solution at a dose of 75 g, which is the maximum dose following the American Diabetes Association (ADA). Then, blood samples (3 mL) were drawn from the median cubital vein or cephalic vein, or basilic vein a put in sodium fluoride tubes at 0, 30, 60, and 120 min after glucose loading to determine plasma glucose levels.

At day 1 after taking the KP capsules, another 3 mL of blood were drawn to an EDTA tube at 0, 15, 30, and 45 min, and 1, 2, 4, 6, 8, 10, and 12 h for pharmacokinetic study. The blood samples were frozen at −20 °C until analysis. The methoxyflavone concentrations in blood samples were further analyzed by using HPLC (Figure 1).

### 2.3. Analysis of Blood Biochemical Parameters

The blood samples were analyzed for complete blood count, electrolytes, glucose, lipid profile, and liver and kidney functions by the Clinical Chemistry Unit at the Faculty of Associated Medicine, Khon Kaen University, using an automated clinical analyzer (Hitachi 912, Tokyo, Japan).

### 2.4. Quantization of Methoxyflavones by HPLC Method

#### 2.4.1. Sample Preparation

The blood sample (2.5 mL) was transferred to a 15 mL tube, and then spiked with 10 µL 17-α-hydroxyprogesterone (55 µg/mL) as internal standard. Six mL of acetonitrile was added as an extraction solvent, mixed for 1 min, and sonicated for 10 min, then centrifuged at 1000 rpm for 10 min. The organic solvent layer was collected with three time repeats in a crucible. The solvents were pooled and evaporated to dryness at room temperature. Then, the residue was reconstituted with 1 mL of methanol and further injected into the HPLC system.

Methoxyflavone concentration in the KP capsule was also determined by dissolving the content of the capsule in methanol and filtration through a 0.45 µm syringe filter; then the filtrate was injected to HPLC.

#### 2.4.2. HPLC System

Methoxyflavones were analyzed by Agilent^®^ 1200 series HPLC (Waldbronn, Germany) with UV detector using an Agilent^®^ Hypersil ODS column (5 µm, 4.6 × 250 mm, Waldbronn, Germany) as the stationary phase, set at 55 °C. A mixture of 0.2% orthophosphoric acid in water (mobile phase A) and methanol (mobile phase B) was used [38]. Briefly, the mobile phase was used as gradient elution with a linear gradient from 40% of mobile phase A and 60% of mobile phase B at a flow rate of 1.2 mL/min at 0 min, 47% of mobile phase A and 53% of mobile phase B at a flow rate of 1 mL/min at 5 min, 60% of mobile phase A and 40% of mobile phase B at a flow rate of 0.7 mL/min at 40 min, and 44% of mobile phase A and 56% of mobile phase B at a flow rate of 1 mL/min at 80 min. The injection volume was 20 µL. The wavelength of UV detection (Agilent^®^ Waldbronn, Germany) was set at 254 nm.

### 2.5. Data Analysis

Area under the blood concentration and time profiles (AUC, ng∙h/mL) was calculated by using the Phoenix WinNonlin program (Pharsight, Mountain View, CA, USA) by noncompartment methods. The maximum concentration (C_max_, ng/mL) and time to C_max_ (T_max_, h) were obtained directly from the blood concentration and time profiles. Elimination rate constant (Ke, h^−1^) was obtained from the slope of log-linear regression of elimination range of blood concentration-time profile. Half-life (T_1/2_, h) was calculated following the equation 0.693/Ke. Statistical analyses were performed using SPSS version 17.0 (SPSS, Inc., Chicago, IL, USA). The Kolmogorov-Smirnov test was used to determine the normal distribution of the data. The data were normally distributed. Paired *t*-test and one-way ANOVA with LSD multiple comparison were used for testing of significant difference within and between groups, respectively; *p*-values less than 0.05 were considered statistically significant.

## 3. Results

### 3.1. Subject Characteristics

From 45 volunteers that fulfilled the inclusion/exclusion criteria, one subject was lost at day 28 (Figure 2). The volunteers were generally healthy, there was no clinically significant finding of hematology and clinical chemistry, and there was an absence of chronic disease. The placebo group included 7 male and 8 female subjects. Six male and 9 female subjects were in the KP extract 90 mg/day group. The volunteers who received 180 mg/day of KP extract were 8 males and 6 females. The general characteristics of the volunteers were as followed: the age ranged from 25 to 28 years old and their weights were from 54 to 58 kg with BMI levels around 20–22 kg/m^2^ (Appendix A). Blood biochemical parameters of all subjects were in normal ranges. No significant differences of all parameters among three groups were detected. Compliance of all volunteers ranged from 96 to 100% within 28 days of treatment. The incidences of adverse events throughout 28 days of treatment were as follows: one volunteer from the placebo group experienced constipation in the 1st week and recovered in the 2nd week; one volunteer from the KP 90 mg/day group experienced a hunger throughout the 28 days of administration. Nausea, vomiting, and anorexia were found in one volunteer who received 180 mg of KP extract at the 2nd and 3rd week, although it recovered by the 4th week. No severe adverse events were reported in this study. In addition, the parameters of blood hematology, biochemistry, liver function, and kidney function of the all volunteers enrolled in this study for 28 days were within normal limits and there was no clinically important appearance (Appendix A).

### 3.2. Oral Glucose Tolerance Test

As shown in Table 2, after the glucose loading, the mean plasma glucose level of the placebo group was continuously increasing from 82.6 ± 7.0 mg/dL at the initial time (0 min) to be 134.7 ± 23.5 at 30 min, then gradually decreased to be 120.9 ± 37.9 mg/dL at 60 min and came back to the normal level at 120 min of the treatment. Similar results were observed in both groups of KP extract treatments. The pattern of blood glucose changes at 30 and 60 min within group of each treatment was statistical different (*p*-value < 0.05). When compared between groups, there were no statistically significant differences of blood glucose levels among placebo, 90, and 180 mg/day of KP extract-treated groups (*p* > 0.05). The results of the oral glucose tolerance test after 28 days of KP extract treatment are shown in Table 3. The significant changes of mean plasma glucose levels at every time points within group of placebo, 90, and 180 mg/day of KP extract-treated groups were similar to those observed at the baseline (*p*-value < 0.05). For between group comparison, there were no statistically significant differences in the blood glucose pattern among these three treatments (*p*-value > 0.05). In order to detect the post-intervention status, the area under the curve (AUC) of blood glucose levels of volunteers after glucose loading at 0–120 min of baseline and 28 days of KP treatment were calculated (Table 4). An administration of KP extract at both 90 and 180 mg/day doses for 28 days did not affect the blood glucose level under the glucose tolerance test in healthy volunteers and detected that there were no statistically significant differences of AUC values of the baseline and 28 days after the KP treatment (*p*-value > 0.05). This finding suggests the safety effect of KP extract consumption on normal blood glucose level. 

### 3.3. Pharmacokinetics

Chromatograms of standard methoxyflavones, KP capsule, and the blood methoxyflavones are shown in Figure 1. The validation method of the HPLC system for determination of methoxyflavones was successful with a correlation coefficient of PMF and TMF at R^2^ 0.98. Limit of detection (LOD) and limit of quantitation (LOQ) of TMF were 0.002 and 0.005 µg/mL, respectively. LOD and LOQ of PMF were 0.003 and 0.007 µg/mL, respectively. Intra-day and inter-day precision of PMF and TMF ranged 0.1 to 1.6 %RSD. Recovery of PMF and TMF ranged from 95.6 to 98.3% and 96.5 to 100.7%, respectively. Amounts of PMF and TMF in the KP capsule in this study were 38.37 and 29.75 mg/g of powder in capsule, respectively. Therefore, the KP capsule contained PMF and TMF at 13.43 and 10.41 mg/capsule, respectively.

After treatment with 90 and 180 mg/day of KP extract, the blood samples of each subject were drawn at various times for 12 h and analyzed for DMF, TMF, and PMF by the HPLC method. The extraction efficacy of the spiked methoxyflavones in blood was more than 75% recovery. PMF and TMF concentrations could be detected in a certain number of subjects. Only 9 of 14 subjects receiving 180 mg/day of KP extract gave the complete data for the blood PMF profile with time point intervals, whereas blood TMF of four subjects could be determined. In the case of subjects receiving 90 mg/day of KP extract, blood PMF could be determined in only nine subjects. As illustrated in Figure 3, the human blood concentrations of PMF reached a peak of 26.0 ± 4.7 ng/mL at 4.02 ± 0.54 h and 71.2 ± 11.3 ng/mL at 4.02 ± 0.37 h after taking KP extracts of 90 mg and 180 mg, respectively. PMF levels in the blood of both initial dosage levels reached zero concentration at 10 h. TMF levels in blood samples could be detected only at a dose of 180 mg/day of KP extract, which displayed C_max_ (63.0 ± 18.0 ng/mL) at 6.03 ± 0.96 h. At the final point of blood drawing (12 h), TMF was still detected in blood at a concentration of 20–25 ng/mL. TMF concentrations in blood samples at dose of 90 mg/day of KP extract were lower than the limit of quantitation. Therefore, the pharmacokinetics of TMF of 90 mg/day dose of KP extract were not able to be reported. All pharmacokinetic parameters of both doses of KP extract are shown in Table 5. Ke of PMF and TMF ranged from 0.32 ± 0.13 to 0.75 ± 0.26 h^−1^. Half-lives of both of PMF and TMF ranging between 1.83 and 3.30 h. Vd values of PMF in volunteers who received 90 and 180 mg/day of KP extract were 218.1 ± 99.3 and 199.0 ± 48.6 L, respectively. At a dose of 180 mg/day of KD extract, the Vd value of TMF (88.3 ± 24.3 L) was lower than that of PMF (199.0 ± 48.6 L). Cl values of PMF at doses of 90 and 180 mg/day and for TMF at the dose of 180 mg/day of KP extract were 57.8 ± 20.8, 54.7 ± 11.2, and 13.0 ± 6.6 L/h, respectively.

## 4. Discussion

To our knowledge, this is the first report on the oral glucose tolerance test and pharmacokinetic study of KP extract in healthy volunteers. KP has centuries of traditional uses for longevity promotion, anti-fatigue, and providing energy expenditure. It is safe and efficient and there are currently several KP products on the market as food and sport supplements. Our preliminary study in rats showed that a KP extract administered at 150 and 300 mg/kg BW decreased blood glucose levels in rat after treatment for 4 weeks (unpublished data). The safety factor was 100. Body weight was 60 kg. Therefore, the dose in this study was KP extract 90 and 180 mg/kg BW. Duration of treatment in this study was designed at 28 days. Moreover, our initial clinical trials on KP extract have demonstrated that at 180 mg/day dose, it significantly enhanced physical fitness, increased muscle strength, and improved aerobic capacity in soccer players [33], and KP extract at a dose of 90 mg/day could enhance the physical fitness in healthy elderly volunteers [32]. Sport supplement formulation containing the combination of KP extract with two other plants also increased physical endurance in healthy volunteers [39]. Considering the clinical effects on physical fitness, promotion of energy expenditure, anti-obesity, and the hyperglycemic effect in an animal model [6,26,33,34], the effect of KP extract on glucose tolerance test at doses of 90 and 180 mg/day for one month consecutive consumption was conducted in healthy volunteers in this study. It is interesting to find that KP extract at doses of 90 and 180 mg/day had no effect on the capacity of the body to manage glycemic condition after the passage of glucose into the human gut and kept the normal limit of blood glucose within 2 h as similarly observed in the placebo group. The results suggest that KP extract does not interfere in the process of blood glucose homeostasis in the normal subjects, which is in contrast to a previous report on the hypoglycemic effect of KP extract in streptozocin-induced diabetic animals [30]. It may imply that KP has a differential effect in glucose metabolism in the diabetic status. The mechanism involved may be via glucose uptake or insulin related glucose metabolic pathways that need to be further elucidated. However, our finding on the glucose tolerance test in this study agrees with the report in normal rats [23]. Moreover, the KP extract also decreased blood glucose after glucose loading in obese diabetic mice, which supports the selective effect of KP extract on glucose metabolism in various types of diabetes. To understand the energy enhancement effect of KP, further studies on the other mechanisms of action including glucose uptake, glycolysis/gluconeogenesis, lipolysis/lipogenesis need to be conducted. There are some supportive studies on this matter that reported that KP improved lipid metabolism. KP extract decreased body weight gain and visceral fat accumulation in obese diabetic mice [23]. The reduction of triglyceride level, increment of oxygen consumption, and promotion of energy metabolism were via the expression of uncoupling protein 1 (UCP 1) in brown adipose tissue (BAT) causing the enhancement of thermogenesis in mice [28]. The expression of UCP 1 in BAT causing a decrease of body weight gain, reduction of fat accumulation, and decrease of blood lipid levels were also reported in obese diabetic mice [29]. Clinical trials on the effect of KP in human subjects have also confirmed these findings as observed that a single dose of KP extract capsule affected the enhancement of whole-body energy expenditure and decreased the body fat without effect on food uptake [26]. Moreover, KP extract decreased body weight gain and abdominal fat accumulation in overweight and pre-obese subjects [34]. Taken together, giving the KP extract at both 90 and 180 mg/day doses for 28 days did not affect the blood glucose level under glucose tolerance test in healthy volunteers. There were no significant differences of AUC of FPG at 0–120 min between baseline and 28 days of treatment and between the KP extract-treated groups and placebo group, it is suggested that KP extract can keep normal blood glucose level and KP extract may facilitate the available sources of energy expenditure in terms of other mechanism such as the enhancement of lipid metabolism.

The pharmacokinetic data in healthy volunteers are important and necessary for drug development, especially for the dosage design of other clinical studies. To the best of our knowledge, clinical pharmacokinetic study of KP extracts in healthy volunteers has never been investigated. Although three major compounds in KP extracts that possess pharmacological effects were used, only TMF and PMF were detected in some human blood samples. Therefore, TMF and PMF were used as chemical markers in this pharmacokinetic study. A single dose pharmacokinetic trial was performed in 30 healthy subjects as designed for the glucose tolerance test (*n* = 15 each for both 90 and 180 mg/day doses of KP extract). PMF and TMF levels were detected by HPLC in only a certain number of blood samples. Unfortunately, DMF could not be detected in all blood samples according to the limited sensitivity of the system. PMF levels were detectable in 33% and 64% of blood samples of subjects receiving 90 and 180 mg/day of KP extract (*n* = 9), whereas TMF levels could be determined only in four blood samples (29%) of subjects receiving 180 mg/day of KP extract. These may reflect the wide variation of methoxyflavone absorption in the gastrointestinal tract. However, our results can be taken as preliminary evidence to support the notion that there is a certain bioavailability of methoxyflavones of KP extract in human blood enabling them to exert their pharmacological effects in the clinical trials. Even though with the above-mentioned limitations of this study, the obtained pharmacokinetic parameters can provide an insight for the dosage regimen in further clinical studies. AUC and C max values of PMF in this study had a linear relationship with the doses. The noticeable variation in the values of T_max_ between PMF and TMF implied that methoxyflavones may be differentially absorbed and metabolized within the gastrointestinal tract as well. Considering the amount of these two methoxyflavones in the 180 mg/day dose, they are equivalent to 26.86 and 20.83 mg of PMF and TMF, respectively. Even though at this dose, the amount of PMF was higher than that of TMF, the AUC of PMF was lower than that of TMF, whereas their C_max_ values were in a linear relationship with the dose of these methoxyflavones, i.e., the C_max_ value of PMF was higher than that of TMF. The lower value of AUC but higher C_max_ values of PMF may result from the higher Ke and Cl of PMF than that of TMF. Accordingly, TMF, which possessed the lower Ke and Cl, had higher AUC than that of PMF. In addition, a gender factor was also considered in the pharmacokinetics of KP in humans. The detectable amount of PMF and TMF were similar in blood samples of both male and female volunteers. In 90 mg/day of KP extract-treated group, PMF could be determined from nine subjects (four males, five females). In 180 mg/day of KP extract-treated group, both TMF and PMF levels were detected from four subjects (each two males and females) and nine subjects (four males and five females). Although too small number of subjects for statistical analysis, the results suggested the similar pharmacokinetic data from male and female subjects, suggesting that there are no gender differences of pharmacokinetic parameters of KP extract. In comparison to the animal study in rats [36], AUC values of TMF were also higher than those of PMF. In rats, TMF and PMF reached the maximum blood concentration at 0.85 and 1.71 h, respectively, which were faster than those in humans. The half-life of TMF was longer than those of PMF in rats (5.91 and 3.12 h), which were also longer than those in humans (3.30 and 1.83 h), respectively. The elimination of PMF was faster than TMF in both rats and humans. These phenomena imply that there is some degree of similarity in the pharmacokinetics of KP extracts in rats and humans. The obtained data from the pharmacokinetic parameters can be used to guide and to design the appropriate dose regimen for future efficacy studies of KP extracts. Furthermore, this study also confirmed the safety of an administration of KP extracts up to 180 mg/day for 28 days consecutively. Blood biochemical and CBC parameters of all KP extract-treated subjects were in the normal limits. The majority of volunteers (93%) showed good compliance to the capsules of KP extracts used in the study; only three subjects (7%), including one in the placebo group, showed mild complains of nausea, vomiting, and anorexia. Moreover, the capsules of KP extract were previously used in studies on the physical fitness in both soccer players and healthy elderly subjects [32,33].

## 5. Conclusions

In conclusion, this study has demonstrated for the first time the oral glucose tolerance test and pharmacokinetics of healthy male and female volunteers who had taken KP extracts at doses of 90 and 180 mg/day. KP extract does not affect the blood glucose homeostasis and the results suggest that methoxyflavones in KP extract may modulate multi-pathways in both carbohydrate and lipid metabolism to promote their energy expenditure effect. In addition, TMF and PMF can be detected and used as markers in human blood and their pharmacokinetic parameters in human were for the first time revealed.

## Figures and Tables

**Figure 1 nutrients-11-01176-f001:**
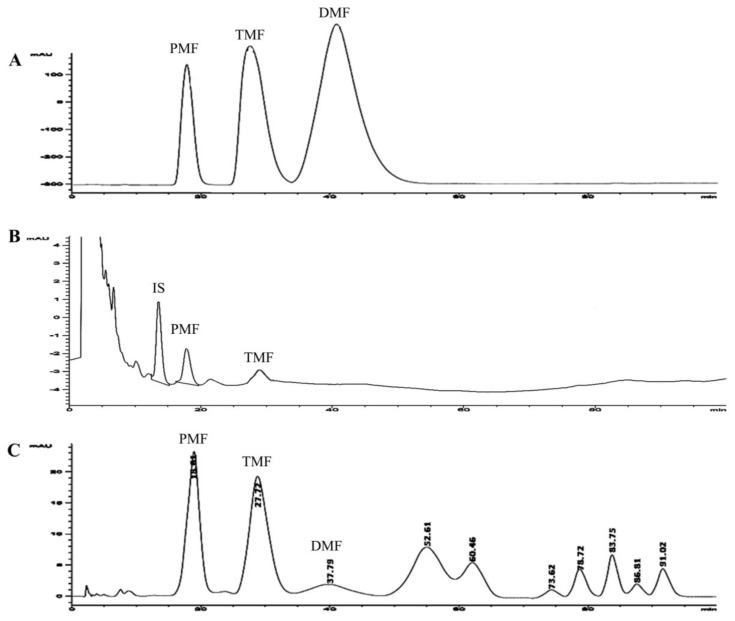
HPLC chromatograms of methoxyflavones; (**A**) = standard compounds; (**B**) = blood sample from a healthy subject following oral *Kaempferia parviflora* Wall. ex Baker (KP) extract 180 mg/day; and (**C**) = KP capsule. PMF: 3,5,7,3′,4′-pentamethoxyflavone, TMF: 5,7,4′-trimethoxyflavone, DMF: 5,7-dimethoxyflavone, IS: internal standard.

**Figure 2 nutrients-11-01176-f002:**
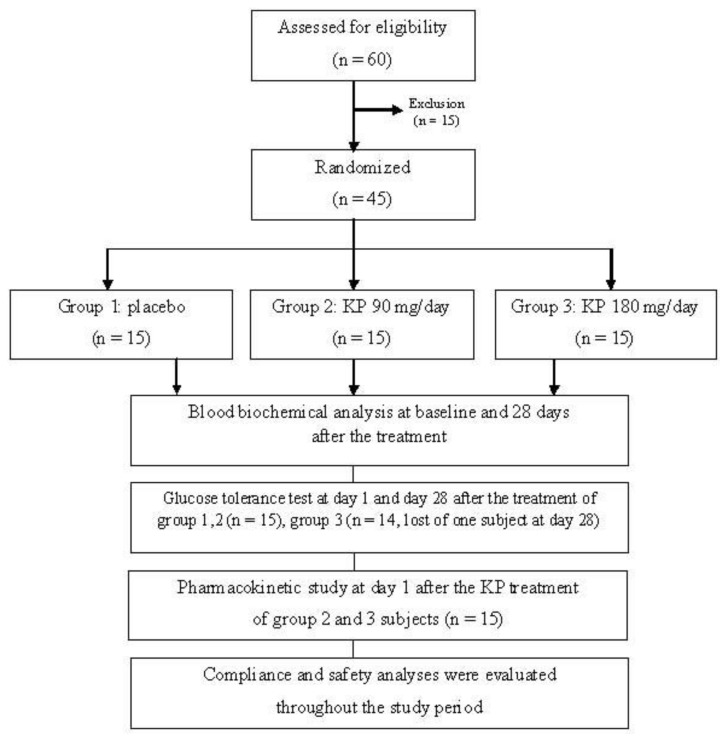
Flowchart illustrating the phases of the study. KP: *Kaempferia parviflora*.

**Figure 3 nutrients-11-01176-f003:**
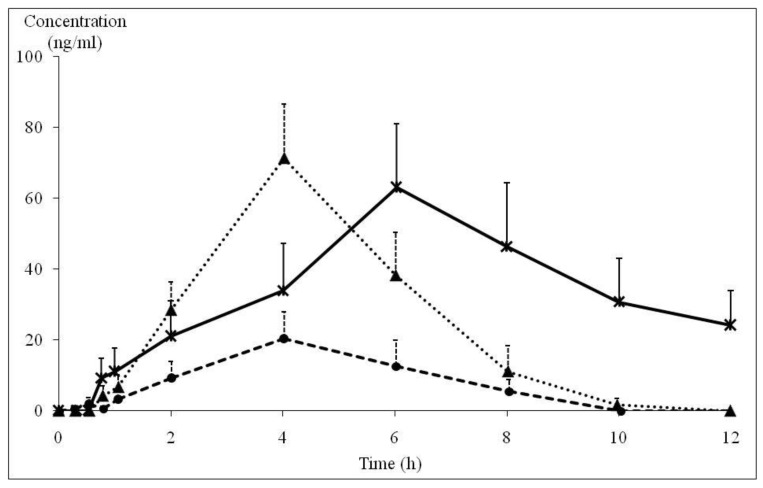
Blood concentration and time profile of PMF and TMF in normal subjects after receiving 90 and 180 mg/day of KP extract (
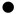
: PMF concentration in KP extract 90 mg/day group (*n* = 5), ▲: PMF concentration in KP extract 180 mg/day group (*n* = 9), 
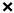
: TMF concentration in KP extract 180 mg/day group (*n* = 4)). PMF: 3,5,7,3′,4′-pentamethoxyflavone, TMF: 5,7,4′-trimethoxyflavone, KP: *Kaempferia parviflora*.

**Table 1 nutrients-11-01176-t001:** Composition of test products.

Composition (mg)	Serving Size: 300 mg/Capsule	
	Placebo	KP Extract
KP extract	0	90
Microcrystalline cellulose	282	192
Silica dioxide	15	15
Magnesium stearate	3	3

KP: *Kaempferia parviflora*.

**Table 2 nutrients-11-01176-t002:** Blood glucose levels of volunteers after glucose loading at baseline.

Group	FPG (mg/dL, mean ± SD)			
	0 min	30 min	60 min	120 min
Placebo (*n* = 15)	82.6 ± 7.0	134.7 ± 23.5	120.9 ± 37.9	83.7 ± 18.8
*p*-value *	-	<0.001	0.002	0.847
KP 90 mg (*n* = 15)	81.9 ± 6.9	128.5 ± 28.4	113.9 ± 39.3	82.5 ± 14.7
*p*-value *	-	<0.001	0.010	0.902
*p*-value **	0.940	0.491	0.432	0.893
KP 180 mg (*n* = 14)	84.7 ± 7.7	131.6 ± 18.9	123.6 ± 30.3	83.0 ± 17.3
*p*-value *	-	<0.001	0.001	0.711
*p*-value **	0.816	0.730	0.764	0.940
*p*-value ***	0.759	0.730	0.278	0.952

* Paired *t*-test (compared with FPG at 0 min). ** *p*-value compared with placebo group using One-way ANOVA with LSD multiple comparison. *** *p*-value compared with KP 90 mg group using One-way ANOVA with LSD multiple comparison. FPG: fasting plasma glucose, KP: *Kaempferia parviflora*.

**Table 3 nutrients-11-01176-t003:** Blood glucose levels of volunteers after glucose loading at 28 days of the treatment.

Group	FPG (mg/dL, mean ± SD)			
	0 min	30 min	60 min	120 min
Placebo (*n* = 15)	86.7 ± 6.9	126.9 ± 25.4	122.0 ± 41.3	79.5 ± 15.2
*p*-value *	-	<0.001	0.004	0.088
KP 90 mg (*n* = 15)	87.5 ± 6.1	120.6 ± 13.7	115.1 ± 31.7	91.3 ± 19.7
*p*-value *	-	<0.001	0.005	0.502
*p*-value **	0.949	0.588	0.549	0.312
KP 180 mg (*n* = 14)	89.2 ± 10.2	142.5 ± 43.9	129.4 ± 61.9	94.2 ± 36.5
*p*-value *	-	<0.001	0.018	0.542
*p*-value **	0.833	0.186	0.533	0.214
*p*-value ***	0.882	0.064	0.226	0.802

* Paired *t*-test (compared with FPG at 0 min). ** *p*-value compared with placebo group using One-way ANOVA with LSD multiple comparison. *** *p*-value compared with KP 90 mg group using One-way ANOVA with LSD multiple comparison. FPG: fasting plasma glucose, KP: *Kaempferia parviflora*.

**Table 4 nutrients-11-01176-t004:** Area under the curve (AUC) of blood glucose levels of volunteers after glucose loading at baseline and 28 days of the treatment.

Group	AUC of FPG at 0–120 min		*p*-Value* (Baseline vs. Day 28)
	Baseline	Day 28	
Placebo	2224.1 ± 1122.0	2255.0 ± 1459.6	0.880
KP 90 mg	2014.0 ± 1309.1	1676.0 ± 678.5	0.281
KP 180 mg	2186.0 ± 1119.7	2220.0 ± 1512.3	0.884
*p*-value ** (Placebo vs. KP 90 mg)	1.000	1.000	
*p*-value ** (Placebo vs. KP 180 mg)	1.000	1.000	
*p*-value ** (KP 90 mg vs. KP 180 mg)	1.000	1.000	

* Within-group comparison using Paired t test. ** between-group comparing using One-way ANOVA with LSD multiple comparison. FPG: fasting plasma glucose, KP: *Kaempferia parviflora*.

**Table 5 nutrients-11-01176-t005:** Pharmacokinetic parameters.

Parameters *	PMF		TMF
	KP Extract 90 mg/day (*n* = 9)	KP Extract 180 mg/day (*n* = 9)	KP Extract 180 mg/day (*n* = 4)
AUC (ng∙h/mL)	86.7 ± 18.8	291.9 ± 48.2	412.2 ± 203.7
Cmax (ng/mL)	26.0 ± 4.7	71.2 ± 11.3	63.0 ± 18.0
Tmax (h)	4.02 ± 0.54	4.02 ± 0.37	6.03 ± 0.96
Ke (h^−^^1^)	0.75 ± 0.26	0.66 ± 0.13	0.32 ± 0.13
T_1/2_ (h)	2.51 ± 0.88	1.83 ± 0.36	3.30 ± 0.96
Vd (L)	218.1 ± 99.3	199.0 ± 48.6	88.3 ± 24.3
Cl (L/h)	57.8 ± 20.8	54.7 ± 11.2	13.0 ± 6.6

***** Values are extract as mean ± SE. PMF: 3,5,7,3′,4′-pentamethoxyflavone, TMF: 5,7,4′-trimethoxyflavone, KP: *Kaempferia parviflora*, AUC: area un the curve, C_max_: maximum concentration, T_max_: time to C_max_, Ke: Elimination rate constant, T_1/2_: Half-life, Vd: volume of distribution, Cl: clearance.

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
