# Peer review of "Glucose Tolerance Test and Pharmacokinetic Study of Kaempferia parviflora Extract in Healthy Subjects"

_nutrients, 2019, doi:10.3390/nu11051176_

Reviewer 1 Report

The abstract needs to be re-written: The is no indication of the rationale, methods are described out of nowhere, oGTT data are not presented.

Introduction of the main text is sufficient.

Methods are described adequately.

Statistics need more information: Was there a test for normal distribution? Which tests were used for comparison of effect size within and between groups?

Results: Please use decimals, where necessary and useful (not for RR or heart rate, where raw data won't provide decimals as well). One decimal only for blood cell parameter, if not measured more accurately. Please use decimals consistently for each parameter (always one decimal for glucose).

Table 3 is labelled as mean + 95%-CI, but sometimes contains mean + SD + 95%-CI and sometimes only mean + 95%-CI. Please clarify and provide consistent data.

Please consistently provide p-values with 3 decimals.

oGTT values need to be compared not only on the basis of baseline and post-interventional status, but on the basis of interventional change, too (deltaDay1-Day28). Otherwise, the authors cannot claim any information on the effect of KP on glycemic state. Also, Glucose-AUCs should be calculated and evaluated cross-sectionally and interventionally.

Discussion: Currently not possible to evaluate - full results needed.

Citations: Please add:

Kaempferia parviflora Ethanol Extract, a Peroxisome Proliferator-Activated Receptor γ Ligand-binding Agonist, Improves Glucose Tolerance and Suppresses Fat Accumulation in Diabetic NSY Mice.

Ochiai M, Takeuchi T, Nozaki T, Ishihara KO, Matsuo T.

J Food Sci. 2019 Feb;84(2):339-348.

Author Response

Thank you for your  very kind and constructive comments in our manuscript. Please see our point-to-point response as followed:

1. The abstract needs to be re-written: The is no indication of the rationale, methods are described out of nowhere, oGTT data are not presented.

Ans. The necessaries data were added in abstract [line 25-26].

2. Statistics need more information: Was there a test for normal distribution? Which tests were used for comparison of effect size within and between groups?

Ans. Kolmogorov-Smirnov test was used for testing the normal distribution as mentioned in Data analysis section [line 168-169].

3. Results: Please use decimals, where necessary and useful (not for RR or heart rate, where raw data won't provide decimals as well). One decimal only for blood cell parameter, if not measured more accurately. Please use decimals consistently for each parameter (always one decimal for glucose).

Ans. The decimal for values of blood cell and biochemical parameters were changed as suggested [Table 2-4].

4. Table 3 is labelled as mean + 95%-CI, but sometimes contains mean + SD + 95%-CI and sometimes only mean + 95%-CI. Please clarify and provide consistent data.

Ans. The data were changed to mean±SD as shown in Table 3-6 and in text.

5. Please consistently provide p-values with 3 decimals.

Ans. The p-values were corrected to be 3 decimals.

6. oGTT values need to be compared not only on the basis of baseline and post-interventional status, but on the basis of interventional change, too (deltaDay1-Day28). Otherwise, the authors cannot claim any information on the effect of KP on glycemic state. Also, Glucose-AUCs should be calculated and evaluated cross-sectionally and interventionally.

Ans. The glucose AUCs were calculated and shown in Table 7.

7. Citations: Please add:

Kaempferia parviflora Ethanol Extract, a Peroxisome Proliferator-Activated Receptor γ Ligand-binding Agonist, Improves Glucose Tolerance and Suppresses Fat Accumulation in Diabetic NSY Mice. Ochiai M, Takeuchi T, Nozaki T, Ishihara KO, Matsuo T. J Food Sci. 2019 Feb;84(2):339-348.

Ans. This citation was added in Introduction.

Reviewer 2 Report

Manuscript nutrients-499010 entitled “Glucose tolerance test and pharmacokinetic study of 2 Kaempferia parviflora (Kp) extract in healthy subjects” reports that administration of Kp extract had no effect on glucose tolerance test. In addition, it demonstrates the pharmacokinetic parameters of methoxyflavones of KP extract in healthy volunteers in an entitled a phase I study. The data suggest Kp extract is safe and may be of benefit for clinical trials using Kp extract as food, sport supplements and health product development. Previous studies have suggested that KP extract and its major constituents confer several health beneficial effects through in vitro and animal studies. Although pharmacokinetic data of KP extract in rats have been reported to our knowledge, data reporting pharmacokinetics in human are not known. This work continues previous work by the corresponding author in this field.

Minor comments:

This is not a phase I study and this should be corrected accordingly in the abstract. Phase 1 trials aim to find out: how much of the drug is safe to give; what are the side effects; PK parameters; efficacy towards clinical/lab variables. The main aim of phase 1 trials is to find out about doses and side effects and that was not the focus here.

It is not clear what was the rationale used to choose 90 and 180mg/day of Kp extract and this information should be added to the “methods” section instead of being mentioned only in the first paragraph of the “discussion”

Spell check: “pregnant” in line 109

The rationale for choosing period of 28 days for Kp administrations should be also discussed.

Please review the OGTT protocol according to the OMS/ADA/IDF consensus.

Results related to blood analysis should be separated from biochemical results in two different tables

Author Response

Thank you for your  very kind and constructive comments in our manuscript. Please see our point-to-point response as followed:

1. This is not a phase I study and this should be corrected accordingly in the abstract. Phase 1 trials aim to find out: how much of the drug is safe to give; what are the side effects; PK parameters; efficacy towards clinical/lab variables. The main aim of phase 1 trials is to find out about doses and side effects and that was not the focus here.

Ans. In Abstract section the “Phase I trial” was deleted.

2. It is not clear what was the rationale used to choose 90 and 180mg/day of KP extract and this information should be added to the “methods” section instead of being mentioned only in the first paragraph of the “discussion”

The rationale for choosing period of 28 days for KP administrations should be also discussed.

Ans. Due to our preliminary study in rats, giving KP extract at doses of 150 and 300 mg/kg BW for 28 days was able to decrease the blood glucose level (unpublished data). The calculated human doses with safety factor of 100 and sixty kilograms of body weight were used in this study, which were at 90 and 180 mg/kg BW. These statements were added in the Method section. This statements were added in the Method section [line 123-129].

3. Spell check: “pregnant” in line 109

Ans. It was corrected.

4. Please review the OGTT protocol according to the OMS/ADA/IDF consensus.

Ans. The dose of glucose for OGTT in this study was 1.75 g/ kg BW. The maximum dose was 75 g following the American Diabetes Association (ADA). However, the body weight of the volunteers in this study were about 55 kg. Therefore, all volunteers drank 75 g of glucose. The method was changed as shown in the Study design section.

6. Results related to blood analysis should be separated from biochemical results in two different tables

Ans. Table 3 was separated to 2 tables including blood analysis (Table 3) and biochemical results (Table 4).

Round  2

Reviewer 1 Report

1. The abstract needs to be re-written: The is no indication of the rationale, methods are described out of nowhere, oGTT data are not presented.

Ans. The necessaries data were added in abstract [line 25-26].

--> has been sufficiently addressed

2. Statistics need more information: Was there a test for normal distribution? Which tests were used for comparison of effect size within and between groups?

Ans. Kolmogorov-Smirnov test was used for testing the normal distribution as mentioned in Data analysis section [line 168-169].

--> has been partly addressed

still unanswered: Which tests were used for comparison of effect size within and between groups?

3. Results: Please use decimals, where necessary and useful (not for RR or heart rate, where raw data won't provide decimals as well). One decimal only for blood cell parameter, if not measured more accurately. Please use decimals consistently for each parameter (always one decimal for glucose).

Ans. The decimal for values of blood cell and biochemical parameters were changed as suggested [Table 2-4].

--> has been sufficiently addressed

4. Table 3 is labelled as mean + 95%-CI, but sometimes contains mean + SD + 95%-CI and sometimes only mean + 95%-CI. Please clarify and provide consistent data.

Ans. The data were changed to mean±SD as shown in Table 3-6 and in text.

--> has been sufficiently addressed

5. Please consistently provide p-values with 3 decimals.

Ans. The p-values were corrected to be 3 decimals.

--> has been sufficiently addressed

6. oGTT values need to be compared not only on the basis of baseline and post-interventional status, but on the basis of interventional change, too (deltaDay1-Day28). Otherwise, the authors cannot claim any information on the effect of KP on glycemic state. Also, Glucose-AUCs should be calculated and evaluated cross-sectionally and interventionally.

Ans. The glucose AUCs were calculated and shown in Table 7.

--> has been partly addressed

still unanswered: oGTT values need to be compared not only on the basis of baseline (Tbl. 5) and post-interventional (Tbl. 6) status, but on the basis of interventional change, too (delta from Day1 to Day28). Otherwise, the authors cannot claim any information on the interventional effect of KP on glycemic state compared to placebo. KP90 obviously leads to a considerable change of Glucose-AUC (see table 7). Comparison of changes needs to be done for Glu0, Glu30, Glu60, Glu120 and Glu-AUC.

7. Citations: Please add:

Kaempferia parviflora Ethanol Extract, a Peroxisome Proliferator-Activated Receptor γ Ligand-binding Agonist, Improves Glucose Tolerance and Suppresses Fat Accumulation in Diabetic NSY Mice. Ochiai M, Takeuchi T, Nozaki T, Ishihara KO, Matsuo T. J Food Sci. 2019 Feb;84(2):339-348.

Ans. This citation was added in Introduction.

--> has been sufficiently addressed

Author Response

Response to reviewer:

1. Statistics need more information: Was there a test for normal distribution? Which tests were used for comparison of effect size within and between groups?

still unanswered: Which tests were used for comparison of effect size within and between groups?

Ans. Kolmogorov-Smirnov test was used for testing the normal distribution as mentioned in Data analysis section [line 170-171].

--> has been partly addressed

still unanswered: Which tests were used for comparison of effect size within and between groups?

Ans.     The data were normal distribution. Paired t-test was used for testing of significant difference within group. One-way ANOVA with LSD multiple comparison was used for testing of significant difference between group. This information was added [line 171-172].

To explain for the sample size calculation, the power of 80%, alpha (α) of 0.05, and standardized effect size of 1.05 (which is a large effect size) were used as followed:

            When, Za/2 is z-score corresponding to confidence level of 95%, Zb is obtained form level of power at 80%, and D is standardized effect size.

Therefore, the sample size in this study was 15 per group.

            The statement, “The sample size was 15 per group which was calculated by using power of 80%, alpha of 0.05, and standardized effect size of 1.05.” was added (line 109-110).

2. oGTT values need to be compared not only on the basis of baseline and post-interventional status, but on the basis of interventional change, too (deltaDay1-Day28). Otherwise, the authors cannot claim any information on the effect of KP on glycemic state. Also, Glucose-AUCs should be calculated and evaluated cross-sectionally and interventionally.

still unanswered: oGTT values need to be compared not only on the basis of baseline (Tbl. 5) and post-interventional (Tbl. 6) status, but on the basis of interventional change, too (delta from Day1 to Day28). Otherwise, the authors cannot claim any information on the interventional effect of KP on glycemic state compared to placebo. KP90 obviously leads to a considerable change of Glucose-AUC (see table 7). Comparison of changes needs to be done for Glu0, Glu30, Glu60, Glu120 and Glu-AUC.

Ans. The glucose AUCs were calculated and shown in Table 7.

--> has been partly addressed

still unanswered: oGTT values need to be compared not only on the basis of baseline (Tbl. 5) and post-interventional (Tbl. 6) status, but on the basis of interventional change, too (delta from Day1 to Day28). Otherwise, the authors cannot claim any information on the interventional effect of KP on glycemic state compared to placebo. KP90 obviously leads to a considerable change of Glucose-AUC (see table 7). Comparison of changes needs to be done for Glu0, Glu30, Glu60, Glu120 and Glu-AUC.

Ans.

The paragraph "3.2 Oral glucose tolerance test" was re-written as followed: As shown in Table 5, after the glucose loading, the mean plasma glucose level of placebo group was continuously increasing from 82.6±7.0 mg/dl at the initial time (0 min) to be 134.7±23.5 at 30 min, then gradually decreased to be 120.9±37.9 mg/dl at 60 min and came back to the
normal level at 120 min of the treatment. Similar results were observed in both groups of KP extract treatments. The pattern of blood glucose changes at 30 and 60 min within group of each treatment was statistical different (p-value<0.05). When compared betweengroups, there were no statistically significant differences of blood glucose levels among placebo, 90 and 180 mg/ day of KP extract-treated groups (p>0.05). The results of oral glucose tolerance test after 28 days of KP extract treatment are shown in Table 6. The significant changes of mean plasma glucose levels at every time points within group of placebo, 90 and 180 mg/day of KP extract-treated groups were similar to those observed at the baseline (p<0.05). For between group comparison,there were no statistically significant differences in the blood glucose pattern among these three treatments (p>0.05). In order to detect the post-intervention status, the area under the curve (AUC) of blood glucose levels of volunteers after glucose loading at 0-120 min of baseline and 28 days of KP treatment were calculated (Table 7). An administration of KP extract at both 90 and 180 mg/day doses for 28 days did not affect the blood glucose level under glucose tolerance test in healthy volunteers as detected that there were no statistically significant differences of AUC values of baseline and 28 days after the KP treatment (p>0.05). This finding suggests the safety effect of KP extract consumption on normal blood glucose level (line 208-226).
